# NEURAL NONNEGATIVE CP DECOMPOSITION FOR HIERARCHICAL TENSOR ANALYSIS

## ABSTRACT

There is a significant demand for topic modeling on large-scale data with complex multi-modal structure in applications such as multi-layer network analysis, temporal document classification, and video data analysis; frequently this multi-modal data has latent hierarchical structure. We propose a new hierarchical nonnegative CANDECOMP/PARAFAC (CP) decomposition (hierarchical NCPD) model and a training method, Neural NCPD, for performing hierarchical topic modeling on multi-modal tensor data. Neural NCPD utilizes a neural network architecture and backpropagation to mitigate error propagation through hierarchical NCPD.

## 1 INTRODUCTION

The recent explosion in the collection and availability of data has led to an unprecedented demand for scalable data analysis techniques. Furthermore, data that has a multi-modal tensor format has become ubiquitous across numerous fields (Cichocki et al., 2009). The need to reduce redundant dimensions (across modes) and to identify meaningful latent trends within data has rightly become an integral focus of research within signal processing and computer science. An important application of these dimension-reduction techniques is *topic modeling*, the task of identifying latent topics and themes of a dataset in an unsupervised or partially supervised approach. A popular topic modeling approach for matrix data is the dimension-reduction technique nonnegative matrix factorization (NMF) (Lee & Seung, 1999), which is generalized to multi-modal tensor data by the nonnegative CP decomposition (NCPD) (Carroll & Chang, 1970; Harshman et al., 1970). These models identify $r$ latent *topics* within the data; here the *rank $r$* is a user-defined parameter that can be challenging to select without *a priori* knowledge or a heuristic selection procedure.

In topic modeling applications, one often additionally wishes to understand the hierarchical topic structure (i.e., how the topics are naturally related and combine into supertopics). For matrices (tensors), a naive approach is to apply NMF (NCPD) first with rank $r$ and then again with rank $j < r$, and simply identify the $j$ supertopics as linear (multilinear) combinations of the original $r$ subtopics. However, due to the nonconvexity of the NMF (NCPD) objective function, the supertopics identified in this way need not be linearly (multi-linearly) related to the subtopics. For this reason, hierarchical models which enforce these relationships between subtopics and supertopics have become a popular direction of research. A challenge of these models is that the nonconvexity of the model at each level of hierarchy can yield cascading error through the layers of models; several works have proposed techniques for mitigating this cascade of error (Flenner & Hunter, 2018; Trigeorgis et al., 2016; Le Roux et al., 2015; Sun et al., 2017; Gao et al., 2019).

In this work, we propose a hierarchical NCPD model and Neural NCPD, an algorithm for training this model which exploits backpropagation techniques to mitigate the effects of error introduced at earlier (subtopic) layers of hierarchy propagating downstream to later (supertopic) layers. This approach allows us to (1) explore the topics learned at different ranks simultaneously, and (2) illustrate the *hierarchical* relationship of topics learned at different tensor decomposition ranks.

**Notation.** We follow the notational conventions of Goodfellow et al. (2016); e.g., tensor $\mathbf{X}$, matrix $X$, vector $\boldsymbol{x}$, and (integer or real) scalar $x$. In all models, we use variable $r$ (with superscripts denoting layer of hierarchical models) to denote model rank and use $j$ when indexing through rank-one

components. In all tensor decomposition models, we use $k$ to denote the order (number of modes) of the tensor and use $i$ when indexing through modes of the tensor. In all hierarchical models, we use $\mathcal{L}$ to denote the number of layers in the model and use $\ell$ to index layers. We let $\otimes$ denote the vector outer product and adopt the CP decomposition notation

$$[\![\boldsymbol{X}_1, \boldsymbol{X}_2, \cdots, \boldsymbol{X}_k]\!] \equiv \sum_{j=1}^{r} \boldsymbol{x}_j^{(1)} \otimes \boldsymbol{x}_j^{(2)} \otimes \cdots \otimes \boldsymbol{x}_j^{(k)}, \tag{1}$$

where $\boldsymbol{x}_j^{(i)}$ is the $j$th column of the $i$th factor matrix $\boldsymbol{X}_i$ (Kolda & Bader, 2009).

**Contributions.** Our main contributions are two-fold. First, we propose a novel hierarchical nonnegative tensor decomposition model that we denote *hierarchical NCPD* (HNCPD). Our model treats all tensor modes alike and the output is not affected by the order of the modes in the tensor representation; this is a property not shared by other hierarchical tensor decomposition models such as that of Cichocki et al. (2007a). Second, we propose an effective neural network-inspired training method that we call Neural NCPD. This method builds upon the Neural NMF method proposed in Gao et al. (2019), but is not a direct extension; Neural NCPD consists of a branch of Neural NMF for each tensor mode, but the backpropagation scheme must be adapted for factorization information flow between branches.

**Organization.** In the remainder of Section 1, we present related work on tensor decompositions and training methods. In Section 2, we present our main contributions, hierarchical NCPD and the Neural NCPD method. In Section 3, we test Neural NCPD on real and synthetic data, and offer some brief conclusions in Section 4. We include justification of several computational details of our method and further experimental results in Appendix A.

## 1.1 RELATED WORK

In this section, we introduce NMF, hierarchical NMF, the Neural NMF method, and NCPD, and then summarize some relevant work.

**Nonnegative Matrix Factorization (NMF).** Given a nonnegative matrix $\boldsymbol{X} \in \mathbb{R}_{\geq 0}^{n_1 \times n_2}$, and a desired dimension $r \in \mathbb{N}$, NMF seeks to decompose $\boldsymbol{X}$ into a product of two low-dimensional nonnegative matrices; dictionary matrix $\boldsymbol{A} \in \mathbb{R}_{\geq 0}^{n_1 \times r}$ and representation matrix $\boldsymbol{S} \in \mathbb{R}_{\geq 0}^{r \times n_2}$ so that

$$\boldsymbol{X} \approx \boldsymbol{AS} = \sum_{j=1}^{r} \boldsymbol{a}_j \otimes \boldsymbol{s}_j, \tag{2}$$

where $\boldsymbol{a}_j$ is a column (topic) of $\boldsymbol{A}$ and $\boldsymbol{s}_j$ is a row of $\boldsymbol{S}$. Typically, $r$ is chosen such that $r < \min\{n_1, n_2\}$ to reduce the dimension of the original data matrix or reveal latent themes in the data. Each column of $\boldsymbol{S}$ provides the approximation of the respective column in $\boldsymbol{X}$ in the lower-dimensional space spanned by the columns of $\boldsymbol{A}$. The nonnegativity of the NMF factor matrices yields clear interpretability; thus, NMF has found application in document clustering (Xu et al., 2003; Gaussier & Goutte, 2005; Shahnaz et al., 2006), and image processing and computer vision (Lee & Seung, 1999; Guillamet & Vitria, 2002; Hoyer, 2002), amongst others. Popular training methods include multiplicative updates (Lee & Seung, 1999; 2001; Lee et al., 2009), projected gradient descent (Lin, 2007), and alternating least-squares (Kim et al., 2008; Kim & Park, 2008).

**Hierarchical NMF (HNMF).** HNMF seeks to illuminate hierarchical structure by recursively factorizing the NMF $\boldsymbol{S}$ matrices; see e.g., (Cichocki et al., 2009). We first apply NMF with rank $r^{(0)}$ and then apply NMF with rank $r^{(1)}$ to the $\boldsymbol{S}$ matrix, collecting the $r^{(0)}$ subtopics into $r^{(1)}$ supertopics. HNMF with $\mathcal{L}$ layers approximately factors the data matrix as

$$\boldsymbol{X} \approx \boldsymbol{A}^{(0)}\boldsymbol{S}^{(0)} \approx \boldsymbol{A}^{(0)}\boldsymbol{A}^{(1)}\boldsymbol{S}^{(1)} \approx \cdots \approx \boldsymbol{A}^{(0)}\boldsymbol{A}^{(1)} \cdots \boldsymbol{A}^{(\mathcal{L}-1)}\boldsymbol{S}^{(\mathcal{L}-1)}. \tag{3}$$

Here the $A^{(i)}$ matrix represents how the subtopics at layer $i$ collect into the supertopics at layer $i+1$. Note that as $\mathcal{L}$ increases, the error $\|\boldsymbol{X} - \boldsymbol{A}^{(0)}\boldsymbol{A}^{(1)} \cdots \boldsymbol{A}^{(\mathcal{L}-1)}\boldsymbol{S}^{(\mathcal{L}-1)}\|_F$ necessarily increases as error propagates with each step. As a result, significant error is introduced when $\mathcal{L}$ is large. Choosing $r^{(0)}, r^{(1)}, \cdots, r^{(\mathcal{L}-1)}$ in practice proves difficult as the number of possibilities grow combinatorially.

**Neural NMF (NNMF).** In the previous work of Gao et al. (2019), the authors developed an iterative algorithm for training HNMF that uses backpropagation techniques to mitigate cascading error through the layers. To form this hierarchical factorization, the Neural NMF algorithm uses a neural net architecture. Each layer $\ell$ of the network has weight matrix $\boldsymbol{A}^{(\ell)}$. In the forward propagation step, the network accepts a matrix $\boldsymbol{S}^{(\ell-1)}$, calculates the nonnegative least-squares solution

$$\boldsymbol{S}^{(\ell)} = q(\boldsymbol{A}^{(\ell)}, \boldsymbol{S}^{(\ell-1)}) \equiv \underset{\boldsymbol{S} \geq 0}{\arg\min} \|\boldsymbol{S}^{(\ell-1)} - \boldsymbol{A}^{(\ell)}\boldsymbol{S}\|_F, \tag{4}$$

and sends the matrix $\boldsymbol{S}^{(\ell)}$ to the next layer. In the backpropagation step, the algorithm calculates gradients and updates the weights of the network, which in this case are the $\boldsymbol{A}$ matrices.

**Nonnegative CP Decomposition (NCPD).** The NCPD generalizes NMF to higher-order tensors; specifically, given an order-$k$ tensor $\mathbf{X} \in \mathbb{R}_{\geq 0}^{n_1 \times n_2 \times \cdots \times n_k}$ and a fixed integer $r$, the approximate NCPD of $\mathbf{X}$ seeks $\boldsymbol{X}_1 \in \mathbb{R}_{\geq 0}^{n_1 \times r}, \boldsymbol{X}_2 \in \mathbb{R}_{\geq 0}^{n_2 \times r}, \cdots, \boldsymbol{X}_k \in \mathbb{R}_{\geq 0}^{n_k \times r}$ so that

$$\mathbf{X} \approx [\![\boldsymbol{X}_1, \boldsymbol{X}_2, \cdots, \boldsymbol{X}_k]\!]. \tag{5}$$

The $\boldsymbol{X}_i$ matrices will be referred to as the NCPD factor matrices. A nonnegative approximation with fixed $r$ is obtained by approximately minimizing the reconstruction error between $\mathbf{X}$ and the NCPD reconstruction. This decomposition has found numerous applications in the area of *dynamic topic modeling* where one seeks to discover topic emergence and evolution (Cichocki et al., 2007b; Traoré et al., 2018; Saha & Sindhwani, 2012). Methods for training NMF models can often be generalized to NCPD; for example, multiplicative updates (Welling & Weber, 2001) and alternating least-squares (Kim et al., 2014).

**Other Related Work.** Other works have sought to mitigate error propagation in HNMF models with techniques inspired by neural networks (Trigeorgis et al., 2016; Le Roux et al., 2015; Sun et al., 2017; Flenner & Hunter, 2018). Additionally, previous works have developed hierarchical tensor decomposition models and methods (Vasilescu & Kim, 2019; Song et al., 2013; Grasedyck, 2010). The model most similar to ours is that of Cichocki et al. (2007a), which we refer to as hierarchical nonnegative tensor factorization (HNTF). This model consists of a sequence of NCPDs, where a factor matrix for one mode is held constant, the remaining factor matrices produce the tensor which is decomposed at the second layer, and this decomposition is combined with the fixed matrix from the previous layer. We note that HNTF is dependent upon the ordering of the modes, and specifically which data mode appears first in the representation of the tensor. We refer to 'HNTF-$i$' as HNTF applied to the representation of the tensor where the modes are reordered with mode $i$ first.

## 2 OUR CONTRIBUTIONS

In this section, we present our two main contributions. We first describe the proposed hierarchical NCPD (HNCPD) model, and then propose a training method, Neural NCPD, for the model.

### 2.1 HIERARCHICAL NCPD (HNCPD)

Given an order-$k$ tensor $\mathbf{X} \in \mathbb{R}^{n_1 \times \cdots \times n_k}$, HNCPD consists of an initial rank-$r$ NCPD layer with factor matrices $\boldsymbol{X}_1, \boldsymbol{X}_2, \ldots, \boldsymbol{X}_k$, each with $r$ columns, and an HNMF with ranks $r^{(0)}, r^{(1)}, \cdots, r^{(\mathcal{L}-2)}$ for each of these factors matrices; that is, for each $\boldsymbol{X}_i$ at layer $\ell$, we factorize $\boldsymbol{X}_i$ as

$$\boldsymbol{X}_i \approx \widetilde{\boldsymbol{X}_i} \equiv \boldsymbol{A}_i^{(0)}\boldsymbol{A}_i^{(1)}...\boldsymbol{A}_i^{(\ell-2)}\boldsymbol{S}_i^{(\ell-2)} \tag{6}$$

where $\boldsymbol{A}_i^{(\ell)}$ has $r^{(\ell)}$ columns; see Figure 1 for a visualization. Thus, HNCPD consists of tensor approximations

$$\mathbf{X} \approx [\![\boldsymbol{A}_1^{(0)}...\boldsymbol{A}_1^{(\ell-2)}\boldsymbol{S}_1^{(\ell-2)}, \cdots, \boldsymbol{A}_k^{(0)}...\boldsymbol{A}_k^{(\ell-2)}\boldsymbol{S}_k^{(\ell-2)}]\!]. \tag{7}$$

To access hierarchical structure between tensor topics at each layer, we need to utilize information in the $\boldsymbol{S}_i^{(\ell)}$ matrices for all modes. To simplify this hierarchical structure, we develop an approximation scheme such that the hierarchical topic structure for all modes is given by a single matrix.

For simplicity, we first consider the two layer case. We note that

$$[\![\widetilde{\boldsymbol{X}}_1, \widetilde{\boldsymbol{X}}_2, \cdots, \widetilde{\boldsymbol{X}}_k]\!] = \sum_{1 \leq j_1, j_2, \ldots j_k \leq r^{(0)}} \alpha_{j_1, j_2, \ldots j_k} \left( (\boldsymbol{A}_1^{(0)})_{:, j_1} \otimes (\boldsymbol{A}_2^{(0)})_{:, j_2} \otimes \ldots \otimes (\boldsymbol{A}_k^{(0)})_{:, j_k} \right)$$

(8)

where $\alpha_{j_1, j_2, \ldots j_k} = \sum_{p=1}^{r} (\boldsymbol{S}_1^{(0)})_{j_1, p} (\boldsymbol{S}_2^{(0)})_{j_2, p} \ldots (\boldsymbol{S}_k^{(0)})_{j_k, p}$; we justify this statement in Appendix A. We refer to decomposition summands in (8) where $j_1 = j_2 = \cdots = j_k$ as vector outer products of *same-index* factor matrix topics, and all other summands as vector outer products of *different-index* factor matrix topics. To identify clear hierarchy, we avoid these different-index column outer products.

The approximation scheme computes matrices $\widetilde{\boldsymbol{A}}_i^{(0)}$ whose columns visualize the desired $r^{(0)}$ NCPD topics along each mode while avoiding different-index column outer products in the decomposition. We approximate the summation (8) by replacing all summands that include column $p_2$ of $\boldsymbol{A}_k^{(0)}$ with a single rank-one vector outer product, $(\widetilde{\boldsymbol{A}}_1^{(0)})_{:, p_2} \otimes (\widetilde{\boldsymbol{A}}_2^{(0)})_{:, p_2} \otimes \ldots \otimes (\widetilde{\boldsymbol{A}}_{k-1}^{(0)})_{:, p_2} \otimes (\boldsymbol{A}_k^{(0)})_{:, p_2}$. To minimize error introduced by this approximation, we transform factor matrices $\boldsymbol{A}_i^{(0)}$ for $i \neq k$ to $\widetilde{\boldsymbol{A}}_i^{(0)}$ by collecting into $(\widetilde{\boldsymbol{A}}_i^{(0)})_{:, p_2}$ the ap-

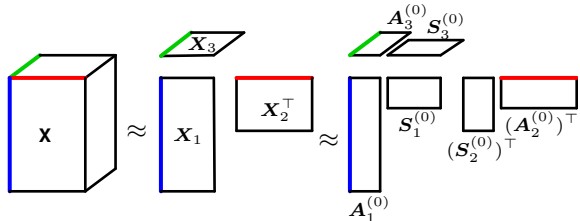

Figure 1: A visualization of a two-layer HNCPD model. The colored edges of the order-three tensor, **X**, represent the three modes.

proximate contribution of all columns of $\boldsymbol{A}_i^{(0)}$ in vector outer products with $(\boldsymbol{A}_k^{(0)})_{:, p_2}$ in (8). That is, for $1 \leq p_1, p_2 \leq r^{(0)}$ and $1 \leq i < k$, let $\boldsymbol{W}_i \in \mathbb{R}^{r^{(0)} \times r^{(0)}}$ be a matrix with

$$(\boldsymbol{W}_i)_{p_1, p_2} = \sum_{j_i = p_1, j_k = p_2, 1 \leq j_1, j_2, \ldots j_k \leq r^{(0)}} \alpha_{j_1, j_2, \ldots, j_k} \quad \text{and} \quad \widetilde{\boldsymbol{A}}_i^{(0)} = \boldsymbol{A}_i^{(0)} \boldsymbol{W}_i, \tag{9}$$

Furthermore, we can identify the topic hierarchy from the $\boldsymbol{S}_k^{(0)}$ matrix. We can generalize this process to later layers $\ell$ by noting that we can group the $\boldsymbol{A}_i$ matrices together, so $\boldsymbol{X}_i \approx \left(\boldsymbol{A}_i^{(0)} \boldsymbol{A}_i^{(1)} \ldots \boldsymbol{A}_i^{(\ell)}\right) \boldsymbol{S}_i^{(\ell)}$. Thus, we can treat this approximation as above, replacing $\boldsymbol{A}_i^{(0)}$ with the product $\boldsymbol{A}_i^{(0)} \boldsymbol{A}_i^{(1)} \ldots \boldsymbol{A}_i^{(\ell)}$.

Like in HNMF, errors in earlier layers can propagate through to later layers and produce highly suboptimal approximations. Challenges encountered during computation of HNMF are exacerbated in an HNCPD model. For this reason, we exploit approaches developed for HNMF in Gao et al. (2019) in our training method Neural NCPD. Furthermore, the computation of HNMF factor matrices for $\boldsymbol{X}_i$ are independent from $\boldsymbol{X}_j$ if the factorizations are applied sequentially; Neural NCPD allows factor matrices in (6) for all other modes to influence the factorization of a given mode.

## 2.2 Neural NCPD

Our iterative method consists of two subroutines, a forward-propagation and a backpropagation. In Algorithms 1 and 2, we display the pseudocode for our proposed method. Following this learning process for the factor matrices in (6), we apply the approximation scheme described in Section 2.1 to the learned factor matrices to visualize the hierarchical structure of the computed HNCPD model.

**Forward Propagation.** The forward-propagation treats $\boldsymbol{A}_i^{(\ell)}$ matrices as neural network weights and uses $\boldsymbol{A}_i^{(\ell)}$ and previous layer output to compute

$$\boldsymbol{S}_i^{(\ell)} = q(\boldsymbol{A}_i^{(\ell)}, \boldsymbol{S}_i^{(\ell-1)}), \tag{10}$$

where $q$ is as defined in equation 4 and $\boldsymbol{S}_i^{(-1)} = \boldsymbol{X}_i$, producing the matrices $\boldsymbol{S}_i^{(0)}, \ldots, \boldsymbol{S}_i^{(\mathcal{L}-2)}$ for $1 \leq i \leq k$. The function $q(\boldsymbol{A}^{(\ell)}, \boldsymbol{S}^{(\ell-1)})$, as a nonnegative least-squares problem, can be

calculated via any convex optimization solver; we utilize an implementation of the Hanson-Lawson algorithm (Lawson & Hanson, 1995). Finally, we pass the $A_i^{(\ell)}$ and $S_i^{(\ell)}$ matrices and **X** into a loss function, which we differentiate and backpropagate.

**Backpropagation.** Our goal is to differentiate our cost function $C$ with respect to the weights in each layer, the $A_i^{(\ell)}$ matrices and backpropagate. This algorithm accepts any first-order optimization method, denoted `optimizer` (e.g., SGD (Robbins & Monro, 1951), Adam (Kingma & Ba, 2014)), but projects the updated weight matrix into the positive orthant to maintain nonnegativity.

For the NCPD task, the most natural loss function is the *reconstruction loss*,

$$C = \|\mathbf{X} - [\![\widetilde{\boldsymbol{X}}_1, \widetilde{\boldsymbol{X}}_2, \cdots, \widetilde{\boldsymbol{X}}_k]\!]\|_F. \tag{11}$$

In order to encourage optimal fit at each layer, we also introduce a loss function that we refer to as *energy loss*. First we denote the approximation of **X** at layer $\ell$ of our network as

---

**Algorithm 1** Forward Propagation

---

**procedure** FORWARDPROP($\{\boldsymbol{X}_i\}_{i=0}^k, \{\boldsymbol{A}_i^{(\ell)}\}_{i=0,\ell=0}^{k,\mathcal{L}-2}$)
    **for** $i = 1, \cdots, k$ **do**
        **for** $\ell = 0, \cdots, \mathcal{L} - 2$ **do**
            $\boldsymbol{S}_i^{(\ell)} \leftarrow q(\boldsymbol{A}_i^{(\ell)}, \boldsymbol{S}_i^{(\ell-1)})$     ▷ see equation 4

---

**Algorithm 2** Neural NCPD

---

**Input:** Tensor $\mathbf{X} \in \mathbb{R}^{n_1 \times n_2 \times \dots \times n_k}$, cost $C$
$\boldsymbol{X}_1, \boldsymbol{X}_2, \dots, \boldsymbol{X}_k \leftarrow$ NCPD($\mathbf{X}$), initialize $\{\boldsymbol{A}_i^{(\ell)}\}_{i=0,\ell=0}^{k,\mathcal{L}-2}$
**for** iterations $= 1, \dots, T$ **do**
    ForwardProp($\{\boldsymbol{X}_i\}_{i=0}^k, \{\boldsymbol{A}_i^{(\ell)}\}_{i=0,\ell=0}^{k,\mathcal{L}-2}$)  ▷ Alg. 1
    **for** $i = 1, \cdots, k, \ \ell = 0, \cdots, \mathcal{L} - 2$ **do**
        $\boldsymbol{A}_i^{(\ell)} \leftarrow \left(\text{optimizer}\left(\boldsymbol{A}_i^{(\ell)}, \frac{\partial C}{\partial \boldsymbol{A}_i^{(\ell)}}\right)\right)^+$
            ▷ any first-order method

---

$$\mathbf{X}_\ell = [\![\boldsymbol{A}_1^{(0)}\boldsymbol{A}_1^{(1)}...\boldsymbol{A}_1^{(\ell-2)}\boldsymbol{S}_1^{(\ell-2)}, \boldsymbol{A}_2^{(0)}\boldsymbol{A}_2^{(1)}...\boldsymbol{A}_2^{(\ell-2)}\boldsymbol{S}_2^{(\ell-2)}, \dots, \boldsymbol{A}_k^{(0)}\boldsymbol{A}_k^{(1)}...\boldsymbol{A}_k^{(\ell-2)}\boldsymbol{S}_k^{(\ell-2)}]\!]. \tag{12}$$

Then, we calculate energy loss as

$$E = \|\mathbf{X} - [\![\boldsymbol{X}_1, \boldsymbol{X}_2, \cdots, \boldsymbol{X}_k]\!]\|_F + \sum_{\ell=0}^{\mathcal{L}-2} \|\mathbf{X} - \mathbf{X}_\ell\|_F. \tag{13}$$

The derivatives of $q(\boldsymbol{A}, \boldsymbol{X})$ with respect to $\boldsymbol{A}$ and $\boldsymbol{X}$ are derived and exploited to differentiate a generic cost function for the hierarchical NMF model in Gao et al. (2019); here we summarize these derivatives and illustrate how to combine them with simple multilinear algebra for HNCPD.

Gao et al. (2019) show that, if $\left(\frac{\partial C}{\partial \boldsymbol{A}_i^{(\ell_1)}}\right)^{\boldsymbol{S}}$ is the derivative of $C$ with respect to $\boldsymbol{A}_i^{(\ell_1)}$ holding the $\boldsymbol{S}$ matrices constant, then

$$\frac{\partial C}{\partial \boldsymbol{A}_i^{(\ell_1)}} = \left(\frac{\partial C}{\partial \boldsymbol{A}_i^{(\ell_1)}}\right)^{\boldsymbol{S}} + \sum_{\substack{\ell_1 \le \ell_2 \le \mathcal{L}-2 \\ 1 \le j \le r}} \boldsymbol{U}_i^{(\ell_1, \ell_2), j}, \tag{14}$$

where $\boldsymbol{U}_i^{(\ell_1, \ell_2), j}$ relates $C$ to $\boldsymbol{A}_i^{(\ell_1)}$ through $\boldsymbol{S}_i^{(\ell_2)}$ and $\boldsymbol{S}_i^{(\ell_1)}$, is defined column-wise ($j$), and depends upon $\left(\frac{\partial C}{\partial \boldsymbol{S}_i^{(\ell_2)}}\right)^*$, the derivative of $C$ with respect to $\boldsymbol{S}_i^{(\ell_2)}$ holding $\boldsymbol{S}_i^{(\ell_2+1)}, \dots, \boldsymbol{S}_i^{(\mathcal{L}-2)}$ constant. The definition of $\boldsymbol{U}_i^{(\ell_1, \ell_2), j}$ is given in Gao et al. (2019) and utilizes, via the chain-rule, the partial derivative of $q(\boldsymbol{A}_i^\ell, \boldsymbol{S}_i^{\ell-1})$ for all $\ell \in [\ell_1, \ell_2]$.

**Example.** The derivative of the previously defined, or other differentiable cost functions can be calculated using these results of Gao et al. (2019) and some simple multi-linear algebra. As an example, we directly compute the backpropagation step for the reconstruction loss function $C$ given in equation 11. Let $\boldsymbol{X}_{(i)}$ be the mode-$i$ matricized version of **X**, and define

$$\boldsymbol{H}_i = \widetilde{\boldsymbol{X}}_k \odot \dots \odot \widetilde{\boldsymbol{X}}_{i+1} \odot \widetilde{\boldsymbol{X}}_{i-1} \odot \dots \odot \widetilde{\boldsymbol{X}}_1, \tag{15}$$

where $\odot$ denotes the Khatri-Rao product (see e.g., (Kolda & Bader, 2009)). Then we have that

$$\left( \frac{\partial C}{\partial \boldsymbol{A}_i^{(\ell_j)}} \right)^{\boldsymbol{S}} = 2 \left( \boldsymbol{A}_i^{(0)} \boldsymbol{A}_i^{(1)} ... \boldsymbol{A}_i^{(\ell_j-1)} \right)^{\top} \left( \boldsymbol{X}_{(i)} - \widetilde{\boldsymbol{X}}_i \boldsymbol{H}_i^{\top} \right) \boldsymbol{H}_i \left( \boldsymbol{A}_i^{(\ell_{j+1})} ... \boldsymbol{A}_i^{(\mathcal{L}-2)} \boldsymbol{S}_i^{(\mathcal{L}-2)} \right)^{\top},$$
(16)

$$\text{and} \quad \left( \frac{\partial C}{\partial \boldsymbol{S}_i^{(\ell_j)}} \right)^{*} = 2 \left( \boldsymbol{A}_i^{(0)} \boldsymbol{A}_i^{(1)} ... \boldsymbol{A}_i^{(\ell_j)} \right)^{\top} \left( \boldsymbol{X}_{(i)} - \widetilde{\boldsymbol{X}}_i \boldsymbol{H}_i^{\top} \right) \boldsymbol{H}_i.$$
(17)

These derivatives are justified in Appendix A. With equation 14, these derivatives are sufficient to calculate the partial derivative of $C$ with respect to any $\boldsymbol{A}$ matrix.

## 3 EXPERIMENTAL RESULTS

We test Neural NCPD on three datasets: one synthetic, one video, and one collected from Twitter. The synthetic dataset is constructed as a simple block tensor with hierarchical structure. The Twitter dataset consists of tweets from political candidates during the 2016 United States presidential election (Littman et al., 2016). We pull the video, a time-lapse of a forest over the span of one year, from (Solheim). We also compare Neural NCPD to *Standard NCPD*, in which we perform an independent NCPD decomposition at each rank, and to *Standard HNCPD*, in which we perform NCPD first on the full dataset, and apply HNMF to the fixed factor matrices; here we sequentially apply NMF to the factor matrices using multiplicative updates and do not update previous layer factorizations as in Neural NCPD. In all experiments, we use `Tensorly` (Kossaifi et al., 2018) for Standard NCPD calculations and to initialize the NCPD layer of our hierarchical NCPD, and in Neural NCPD we do not backpropagate to this layer as the initialization has usually found a stationary point. We use Energy Loss (Eq. 12) for all experiments to encourage fit at every layer. Because we do not backpropagage to the initial factor matrices, the first term in (Eq. 12) is fixed. For the Twitter and video experiments we use the approximation scheme of Section 2.1 to recover the relationship between the columns of the $\boldsymbol{A}_i^{(\ell)}$ matrices and visualize the $\widetilde{A}_i^{(\ell)}$ matrices.

### 3.1 EXPERIMENT ON SYNTHETIC DATA

We test the Neural NCPD algorithm first using a synthetic dataset. This dataset is a rank seven tensor of size $40 \times 40 \times 40$ with positive noise added to each entry; we generate noise as $n = |g|$ where $g \sim \mathcal{N}(0, \sigma^2)$. To generate this dataset, we begin with the all-zeros tensor and create three large nonoverlapping blocks with value 1, and then overlay each block with either two or three additional blocks with value 3. We display this tensor with two levels of noise at the left of Figure 2; here we plot projections of all tensors (and all approximations) along the third mode; that is, we construct a matrix with entries equal to the largest entries of the mode-three fibers (see e.g., (Kolda & Bader, 2009) for relevant definitions). The projections on the remaining two modes are included in the Appendix A, and are all similar to the third mode.

Table 1: Relative reconstruction loss, $C_{\text{rel}}$, on a synthetic dataset for Neural NCPD, Standard HNCPD, and HNTF with two different levels of noise. We list the loss of the approximation $r^{(1)} = 3$. The results of HNTF are similar for all orderings of the modes, so we list only one.

| Method | $\sigma^2 = 0.05$ | | | $\sigma^2 = 0.5$ | | |
|---|---|---|---|---|---|---|
| | $r = 7$ | $r^{(0)} = 5$ | $r^{(1)} = 3$ | $r = 7$ | $r^{(0)} = 5$ | $r^{(1)} = 3$ |
| Neural NCPD | 0.091 | **0.229** | **0.467** | 0.390 | **0.438** | **0.490** |
| Standard HNCPD | 0.091 | 0.525 | 0.674 | 0.390 | 0.441 | 0.643 |
| HNTF | 0.091 | 0.234 | 0.539 | 0.390 | 0.450 | 0.578 |

We run Neural NCPD, Standard HNCPD, and Chichocki et al. on this synthetic dataset at two different levels of noise with three layers of ranks 7, 5, and 3, and display the results in Figure 2 and Table 1; we present the relative reconstruction loss $C_{\text{rel}} = \|\mathbf{X} - [\![\widetilde{\boldsymbol{X}}_1, \widetilde{\boldsymbol{X}}_2, \cdots, \widetilde{\boldsymbol{X}}_k]\!]\|_F / \|\mathbf{X}\|_F$. For each level of noise, we display the rank 7 approximation shared by all methods, and the rank 5

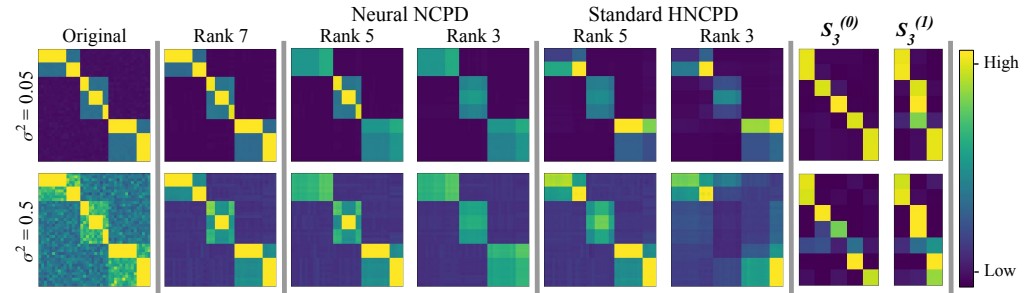

Figure 2: Data tensor **X** with two levels of noise (left), ranks 7, 5, and 3 Neural NCPD and Standard HNCPD approximations of **X** (middle), transposed Neural NCPD $\boldsymbol{S}_3^{(0)}$ and $\boldsymbol{S}_3^{(1)}$ matrices (right).

and rank 3 approximations produced by Neural NCPD and Standard HNCPD. We also display the transposed Neural NCPD $\boldsymbol{S}_3^{(0)}$ and $\boldsymbol{S}_3^{(1)}$ matrices, which show how rank 7 topics collect into ranks 5 and 3 topics. From Table 1, we see that the loss for Neural NCPD is at or below that of Standard HNCPD and HNTF at each rank and level of noise.

## 3.2 TEMPORAL DOCUMENT ANALYSIS

We next apply Neural NCPD to a dataset of tweets from four Republican [R] and four Democratic [D] 2016 presidential primary candidates, (1) Hillary Clinton [D], (2) Tim Kaine [D], (3) Martin O'Malley [D], (4) Bernie Sanders [D], (5) Ted Cruz [R], (6) John Kasich [R], (7) Marco Rubio [R], and (8) Donald Trump [R]; this is constructed from a subset of the dataset of Littman et al. (2016). We use a bag-of-words (12,721 words in corpus) representation of all tweets made by a candidate within bins of 30 days (from February to December 2016), and cap each of these groups at 100 tweets to avoid oversampling from any candidate; resulting in a tensor of size $8 \times 10 \times 12721$.

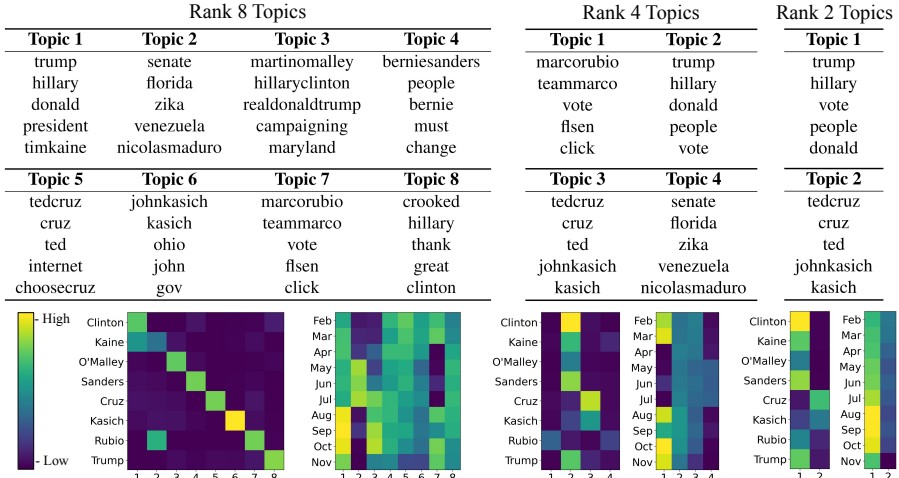

Figure 3: A three-layer Neural NCPD on the Twitter dataset at ranks $r = 8$, $r^{(0)} = 4$ and $r^{(0)} = 2$. At each rank, we display the top keywords and topic heatmaps for candidate and temporal modes.

In Table 2, we display the relative reconstruction loss on the Twitter political dataset for all models. We see that Neural NCPD significantly outperforms Standard HNCPD, slightly outperforms Standard NCPD while offering a hierarchical topic structure, and outperforms all HNTF-$i$, for which loss varies significantly based on the arrangement of the tensor. In Figure 3, we show the topic keywords and factor matrices of a rank 8, 4, and 2 hierarchical NCPD approximation computed by Neural NCPD. Note that in the rank 8 candidates mode factor and keywords we see that nearly every topic is identified with a single candidate. Topic two of the rank 8 approximation aligns with

Table 2: Relative reconstruction loss, $C_{\text{rel}}$, on the Twitter political dataset for Neural NCPD, Standard NCPD, Standard HNCPD, and HNTF at ranks $r = 8$, $r^{(0)} = 4$, and $r^{(1)} = 8$. For HNTF we display the loss given the three possible arrangements of the tensor.

| Method | $r = 8$ | $r^{(0)} = 4$ | $r^{(1)} = 2$ |
|---|---|---|---|
| Neural NCPD | 0.834 | **0.883** | **0.918** |
| Standard NCPD | 0.834 | 0.889 | 0.919 |
| Standard HNCPD | 0.834 | 0.931 | 0.950 |
| HNTF-1 | 0.834 | 0.890 | 0.927 |
| HNTF-2 | 0.834 | 0.909 | 0.956 |
| HNTF-3 | 0.834 | 0.895 | 0.942 |

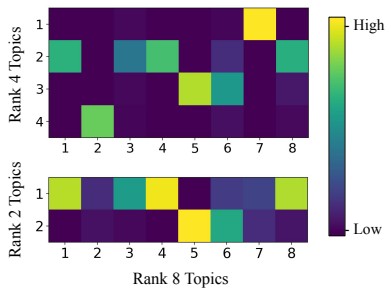

Figure 4: The $\boldsymbol{S}_3^{(0)}$ (top) and $\boldsymbol{S}_3^{(1)}$ (bottom) matrices produced by Neural NCPD on the Twitter dataset.

political issues (the Zika virus and the Venezuelan government) rather than a single candidate, and is temporally most present in May to July 2016 (during the Zika outbreak and the Venezualan state of emergency). Topics one and eight, corresponding to candidates Clinton and Trump, are most present in the months immediately leading up to the election.

At rank 4, we see that topics one and four are inherited from the rank 8 approximation, topic two combines the rank 8 topics of candidates Trump and Clinton (final candidates), and topic 3 combined the topics of candidates Cruz and Kasich (Republicans). Meanwhile, the rank 2 NCPD topics are nearly identical to rank 4 NCPD topics two and three. We display HNTF for each ordering of the tensor modes in Appendix A.

In Figure 4, we display the $\boldsymbol{S}_3^{(0)}$ (top) and $\boldsymbol{S}_3^{(1)}$ (bottom) matrices produced by Neural NCPD on the Twitter dataset, which illustrate how topics collect at each rank. We see topics 5 and 6 from the rank 8 factorization combine to form topic 3 at rank 4 and topic 2 at rank 2. This is expected because both topics include keywords from Cruz and Kasich, who had high presence in topics 5 and 6 respectively in the rank 8 factorization.

In Figure 5, we display the results of performing separate NCPD decomposition of ranks 4 and 2 on the Twitter dataset. We see that the results are similar to those of Neural NCPD, but these independent decompositions lack the clear hierarchical structure provided by Neural NCPD. Note that while the topics corresponding to Kasich and Clinton combine in the rank 4 NCPD, these candidates are present in *different* topics in the rank 2 NCPD; Neural NCPD prevents this breach of hierarchy.

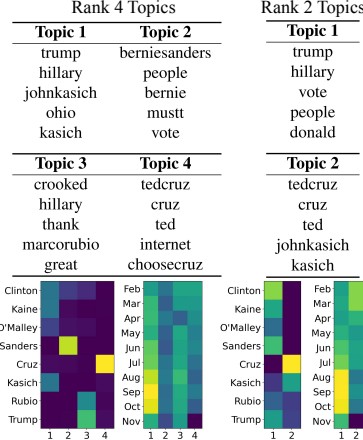

Figure 5: Standard ranks 4 and 2 NCPD of the Twitter dataset. At each rank, we display the top five keywords and candidate and temporal mode heatmaps.

### 3.3 VIDEO DATA ANALYSIS

We next apply Neural NCPD to video data constructed from a year-long time-lapse video of a forest; see Figure 6 for a selection of frames and Figure 11 in Appendix A for more details. We extract 37 frames and flatten each frame (RGB image) into a single matrix, to form a tensor $\mathbf{X}$ of size $37 \times 3 \times 57600$; here the first mode represents frames (temporal mode), the second colors (chromatic mode), and the third pixels (spatial mode).

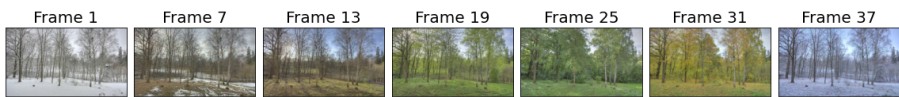

Figure 6: We display seven of 37 extracted frames from a year-long time-lapse video of a forest.

In Figure 7, we show the three-layer Neural NCPD decompositions of the video tensor with ranks 8, 6, and 3. For each rank, we plot the topics in the spatial (left), temporal (top right), and chromatic (bottom right) modes. We note that many of the identified topics represent visual seasonal changes. Topic six of the rank 8 decomposition represents the green and leafy late-summer to early fall. Topic one of the rank 6 decomposition represents the winter sky and leafless trees. Topic three of the rank 3 decomposition represents the summer and fall sky and tree leaves.

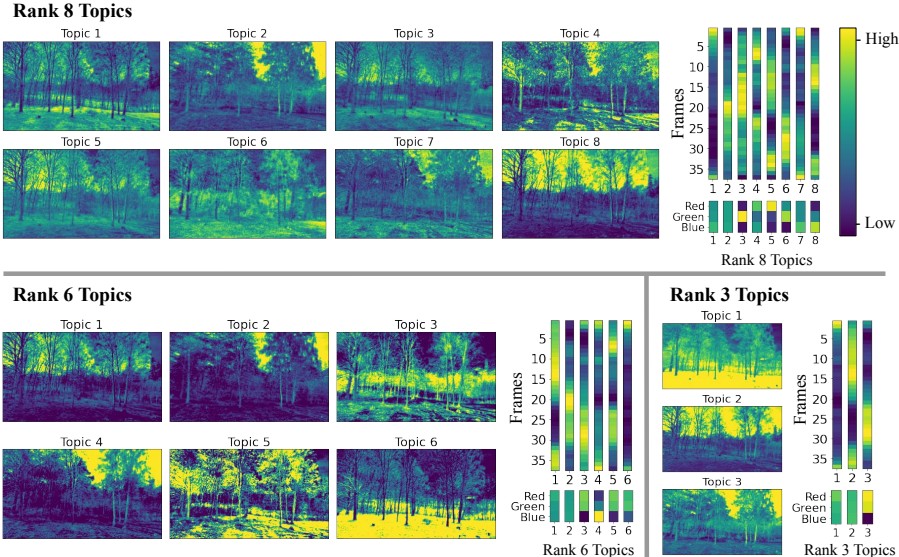

Figure 7: A three-layer Neural NCPD of the time-lapse video at ranks $r = 8$, $r_0 = 6$, and $r_1 = 3$. We display topics at each rank for spatial (left), temporal (top right), and chromatic (bottom right) modes. Relative reconstruction loss is $0.105$, $0.109$, and $0.122$ respectively at each layer.

We additionally apply NMF to the slices of the tensor along a single mode. Slicing along the temporal or spatial modes would make interpretation of the resulting topics challenging, so we choose to slice along the chromatic mode, producing three matrices. In Figure 8, we visualize a rank 3 NMF on each of the three chromatic slices of the video tensor. The chromatic factorizations are nearly identical, illustrating little salient dynamic information. While similar to the rank 3 Neural NCPD layer, the chromatic NMFs obscure much of the chromatic interaction evidenced by Neural NCPD. In particular, Neural NCPD illustrates the spatial and temporal dynamics of *multi-colored* features and their co-occurrence hierarchy, while NMF provides only *single-colored* features and requires far more work to glean multi-colored feature co-occurrence information.

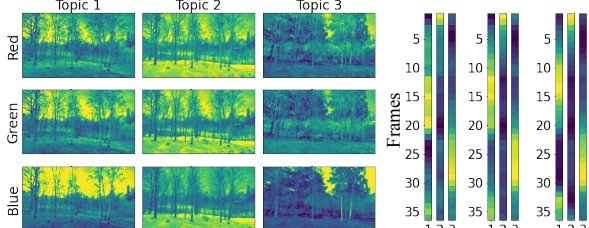

Figure 8: A decomposition of the time-lapse video by rank 3 NMF on slices of the tensor along the chromatic mode. For each color, we display the three topics in spatial (left) and temporal (right) modes. Relative reconstruction loss is $0.101$.

## 4 CONCLUSIONS

In this paper, we introduced the hierarchical NCPD model and presented a novel algorithm, Neural NCPD, to train this decomposition. We empirically demonstrate the promise of this method on both real and synthetic datasets; in particular, this model reveals the hierarchy of topics learned at different NCPD ranks, which is not available to standard NCPD or NMF-based approaches.

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

# A APPENDIX

In this supplementary material, we provide the details of the example derivative computations from Section 2.2, give a justification of the NCPD expansion formula exploited in Section 2.1, and provide further experimental results that we were not able to include in Section 3.

## EXAMPLE DERIVATIONS

Here, we justify the derivations provided in the example in Section 2.2. We note that Anaissi et al. (2020); Kolda & Hong (2019) provide similar derivations for the CP tensor decomposition, but their decompositions do not attempt to further decompose the CP factor matrices, and thus, their results are not sufficient for providing derivatives with respect to the $A$ and $S$ matrices. Consider the full reconstruction loss function for the order-$k$ tensor $\mathbf{X}$,

$$C = \|\mathbf{X} - [\![X_1, X_2, \ldots, X_k]\!]\|_F^2,$$

where for some fixed $1 \le i \le k$, $X_i = ABC$ and consider the gradient $\dfrac{\partial C}{\partial B}$. Let

$$H_i = X_k \odot \ldots \odot X_{i+1} \odot X_{i-1} \odot \ldots \odot X_1.$$

Now, we let $\widetilde{\mathbf{X}} = [\![X_1, X_2, \ldots, X_k]\!]$. Then, if $\widetilde{X}_{(i)}$ is the mode-$i$ matricization of $\widetilde{\mathbf{X}}$ (see e.g., (Kolda & Bader, 2009)), we have

$$\widetilde{X}_{(i)} = X_i(X_k \odot \ldots \odot X_{i+1} \odot X_{i-1} \odot \ldots \odot X_1)^\top] = X_i H_i^\top.$$

Thus, if we let $X_{(i)}$ be the mode-$i$ matricization of $\mathbf{X}$, we have that

$$C = \|X_{(i)} - \widetilde{X}_{(i)}\|_F^2 = \|X_{(i)} - (ABC)(X_k \odot \ldots \odot X_{i+1} \odot X_{i-1} \odot \ldots \odot X_1)^\top\|_F^2.$$

Now, we compute the desired gradient through a series of applications of the chain rule. We then see that

$$\begin{aligned}
\frac{\partial C}{\partial B} &= \frac{\partial C}{\partial X_i H_i^\top} \frac{\partial X_i H_i^\top}{\partial ABC} \frac{\partial ABC}{\partial B} \\
&= A^\top \left( \frac{\partial C}{\partial X_i H_i^\top} \frac{\partial X_i H_i^\top}{\partial X_i} \right) C^\top \\
&= 2A^\top \left( X_{(i)} - X_i H_i^\top \right) H_i C^\top.
\end{aligned}$$

Now, using the calculations above we can proceed in calculating $\dfrac{\partial C}{\partial A_i^{(\ell_j)}}$. Gao et al. (2019) show that

if $\left( \dfrac{\partial C}{\partial A_i^{(\ell_j)}} \right)^S$ denotes derivative of $C$ with respect to $A_i^{(\ell_j)}$, holding the $S$ matrices constant, then we have

$$\frac{\partial C}{\partial A_i^{(\ell_j)}} = \left( \frac{\partial C}{\partial A_i^{(\ell_1)}} \right)^S + \sum_{\substack{\ell_1 \le \ell_2 \le \mathcal{L}-2 \\ 1 \le j \le r}} U_i^{(\ell_1, \ell_2), j}$$

where $U_i^{(\ell_1, \ell_2), j}$ relates $C$ to $A_i^{(\ell_1)}$ through $S_i^{(\ell_2)}$ and $S_i^{(\ell_1)}$, is defined column-wise ($j$), and depends upon $\left( \dfrac{\partial C}{\partial S_i^{(\ell_2)}} \right)^*$, the derivative of $C$ with respect to $S_i^{(\ell_2)}$ holding $S_i^{(\ell_2+1)}, \ldots, S_i^{(\mathcal{L}-2)}$ constant. Thus, $\left( \dfrac{\partial C}{\partial A_i^{(\ell_j)}} \right)^S$ and $\left( \dfrac{\partial C}{\partial S_i^{(\ell_j)}} \right)^*$ are sufficient to calculate $\dfrac{\partial C}{\partial A_i^{(\ell_j)}}$. We calculate the gradient $\left( \dfrac{\partial C}{\partial A_i^{(\ell_j)}} \right)^S$ where

$$C = \|\mathbf{X} - [\![\widetilde{X}_1, \widetilde{X}_2, \ldots, \widetilde{X}_k]\!]\|_F^2$$

and $\widetilde{\boldsymbol{X}}_i = \boldsymbol{A}_i^{(0)} \boldsymbol{A}_i^{(1)} ... \boldsymbol{A}_i^{(\mathcal{L}-2)} \boldsymbol{S}_i^{(\mathcal{L}-2)}$. Since we can assume that $\boldsymbol{A}_i^{(\ell_1)}$ is independent of all other $\boldsymbol{A}$'s and $\boldsymbol{S}$'s, we have that

$$\left( \frac{\partial C}{\partial \boldsymbol{A}_i^{(\ell_j)}} \right)^{\boldsymbol{S}} = 2 \left( \boldsymbol{A}_i^{(0)} \boldsymbol{A}_i^{(1)} ... \boldsymbol{A}_i^{(\ell_j - 1)} \right)^{\top} \left( \boldsymbol{X}_{(i)} - \boldsymbol{X}_i \boldsymbol{H}_i^{\top} \right) \boldsymbol{H}_i \left( \boldsymbol{A}_i^{(\ell_{j+1})} ... \boldsymbol{A}_i^{(\mathcal{L}-2)} \boldsymbol{S}_i^{(\mathcal{L}-2)} \right)^{\top}. \tag{18}$$

Now, we calculate $\left( \frac{\partial C}{\partial \boldsymbol{S}_i^{(\ell_j)}} \right)^{*}$. Since we can assume that $\boldsymbol{S}_i^{(\ell_j)}$ is independent of all other $\boldsymbol{A}$'s and $\boldsymbol{S}$'s, we have that

$$\left( \frac{\partial C}{\partial \boldsymbol{S}_i^{(\ell_j)}} \right)^{*} = 2 \left( \boldsymbol{A}_i^{(0)} \boldsymbol{A}_i^{(1)} ... \boldsymbol{A}_i^{(\ell_j)} \right)^{\top} \left( \boldsymbol{X}_{(i)} - \boldsymbol{X}_i \boldsymbol{H}_i^{\top} \right) \boldsymbol{H}_i. \tag{19}$$

Thus, we have the required derivatives to evaluate $\frac{\partial C}{\partial \boldsymbol{A}_i^{(\ell_j)}}$.

## HNCPD EXPANSION

We now provide brief justification of the expansion of the NCPD in terms of later factorizations used in Section 2.1; that is,

$$[\![ \widetilde{\boldsymbol{X}}_1, \widetilde{\boldsymbol{X}}_2, \cdots, \widetilde{\boldsymbol{X}}_k ]\!] = \sum_{1 \le j_1, j_2, ... j_k \le r^{(0)}} \alpha_{j_1, j_2, ... j_k} \left( (\boldsymbol{A}_1^{(0)})_{:,j_1} \otimes (\boldsymbol{A}_2^{(0)})_{:,j_2} \otimes ... \otimes (\boldsymbol{A}_k^{(0)})_{:,j_k} \right)$$

where $\alpha_{j_1, j_2, ... j_k} = \sum_{p=1}^{r} (\boldsymbol{S}_1^{(0)})_{j_1, p} (\boldsymbol{S}_2^{(0)})_{j_2, p} ... (\boldsymbol{S}_k^{(0)})_{j_k, p}$.

We have that by definition,

$$[\![ \widetilde{\boldsymbol{X}}_1, \widetilde{\boldsymbol{X}}_2, \cdots, \widetilde{\boldsymbol{X}}_k ]\!] = \sum_{p=1}^{r} \left( (\widetilde{\boldsymbol{X}}_1)_{:,p} \otimes (\widetilde{\boldsymbol{X}}_2)_{:,p} \otimes ... \otimes (\widetilde{\boldsymbol{X}}_k)_{:,p} \right).$$

We also have that $\widetilde{\boldsymbol{X}}_i = \boldsymbol{A}_i^{(0)} \boldsymbol{S}_i^{(0)}$ for $1 \le i \le k$, so we have that for each column $p$, $1 \le p \le r$ of $\widetilde{\boldsymbol{X}}_i$,

$$(\widetilde{\boldsymbol{X}}_i)_{:,p} = \sum_{j=1}^{r^{(0)}} (\boldsymbol{S}_i^{(0)})_{j,p} (\boldsymbol{A}_i^{(0)})_{:,j}.$$

Thus, by the linearity of the outer product we have that

$$\left( (\widetilde{\boldsymbol{X}}_1)_{:,p} \otimes (\widetilde{\boldsymbol{X}}_2)_{:,p} \otimes ... (\widetilde{\boldsymbol{X}}_k)_{:,p} \right) = \sum_{1 \le j_1, j_2, ... j_k \le r^{(0)}} \alpha_{p, j_1, j_2, ... j_k} \left( (\boldsymbol{A}_1^{(0)})_{:,j_1} \otimes (\boldsymbol{A}_2^{(0)})_{:,j_2} \otimes ... \otimes (\boldsymbol{A}_k^{(0)})_{:,j_k} \right)$$

where $\alpha_{p, j_1, j_2, ... j_k} = (\boldsymbol{S}_1^{(0)})_{j_1, p} (\boldsymbol{S}_2^{(0)})_{j_2, p} ... (\boldsymbol{S}_k^{(0)})_{j_k, p}$. Now, by noting that

$$\alpha_{j_1, j_2, ..., j_k} = \sum_{p=1}^{r} \alpha_{p, j_1, j_2, ..., j_k}$$

we arrive at the original statement.

## SYNTHETIC EXPERIMENT

In this section, we provide the additional views of the synthetic tensor and computed approximations from Section 3.1. In Figure 2 in the main text, for visualization we displayed the projection of each tensor onto the third mode. In Figure 9, we display the projections of these tensors onto all three modes. We see that due to the simple block structure used to produce the synthetic data tensor, the three modes all tell a similar story; that is, Neural NCPD is able to recover meaningful structure along all three modes.

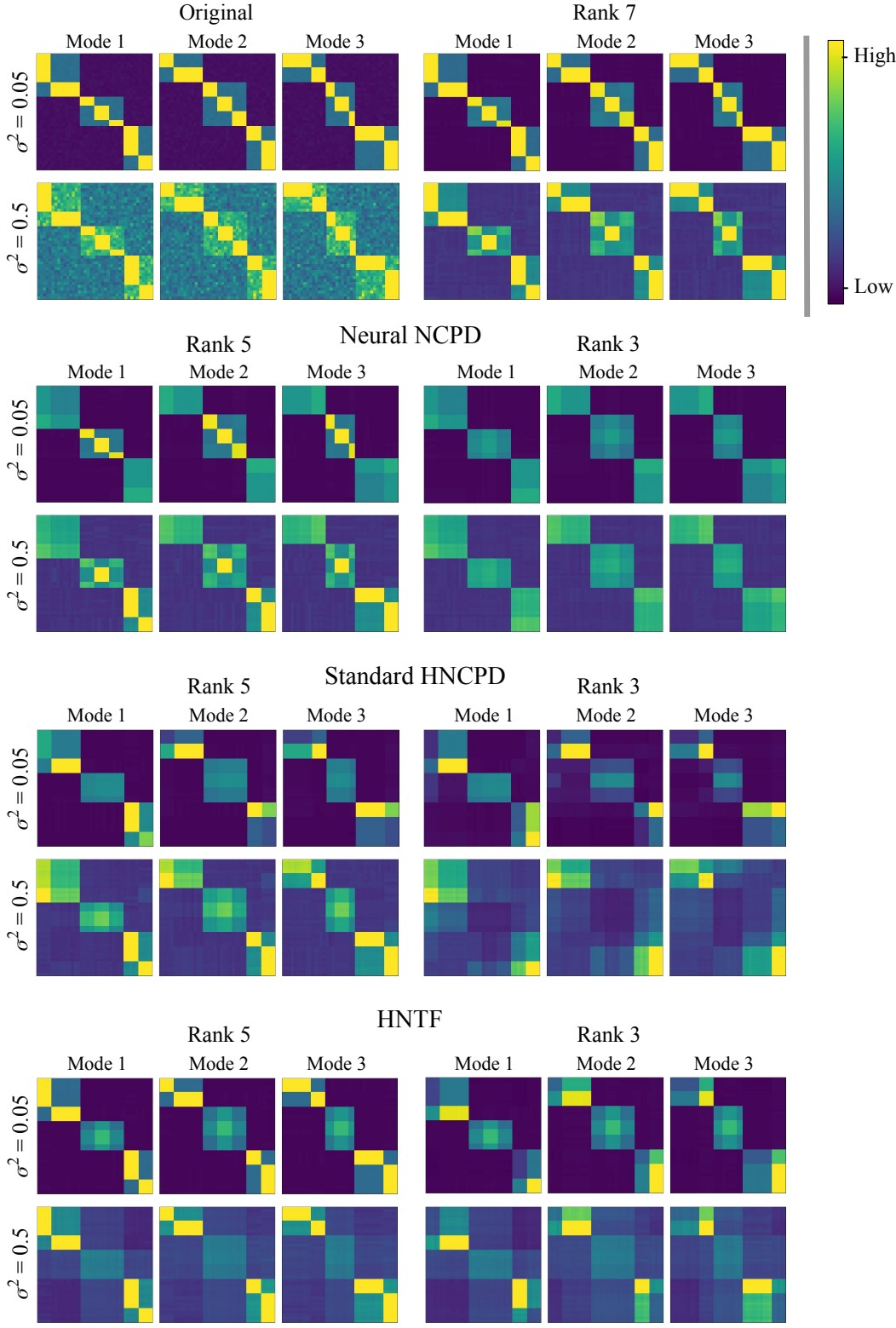

Figure 9: Here we display the projections onto all three modes for the original data tensor **X** and approximations of **X** at ranks $r = 7$, $r^{(0)} = 5$, and $r^{(1)} = 3$ produced by Neural NCPD, Standard HNCPD, and HNTF at two levels of noise.

TEMPORAL DOCUMENT ANALYSIS EXPERIMENT

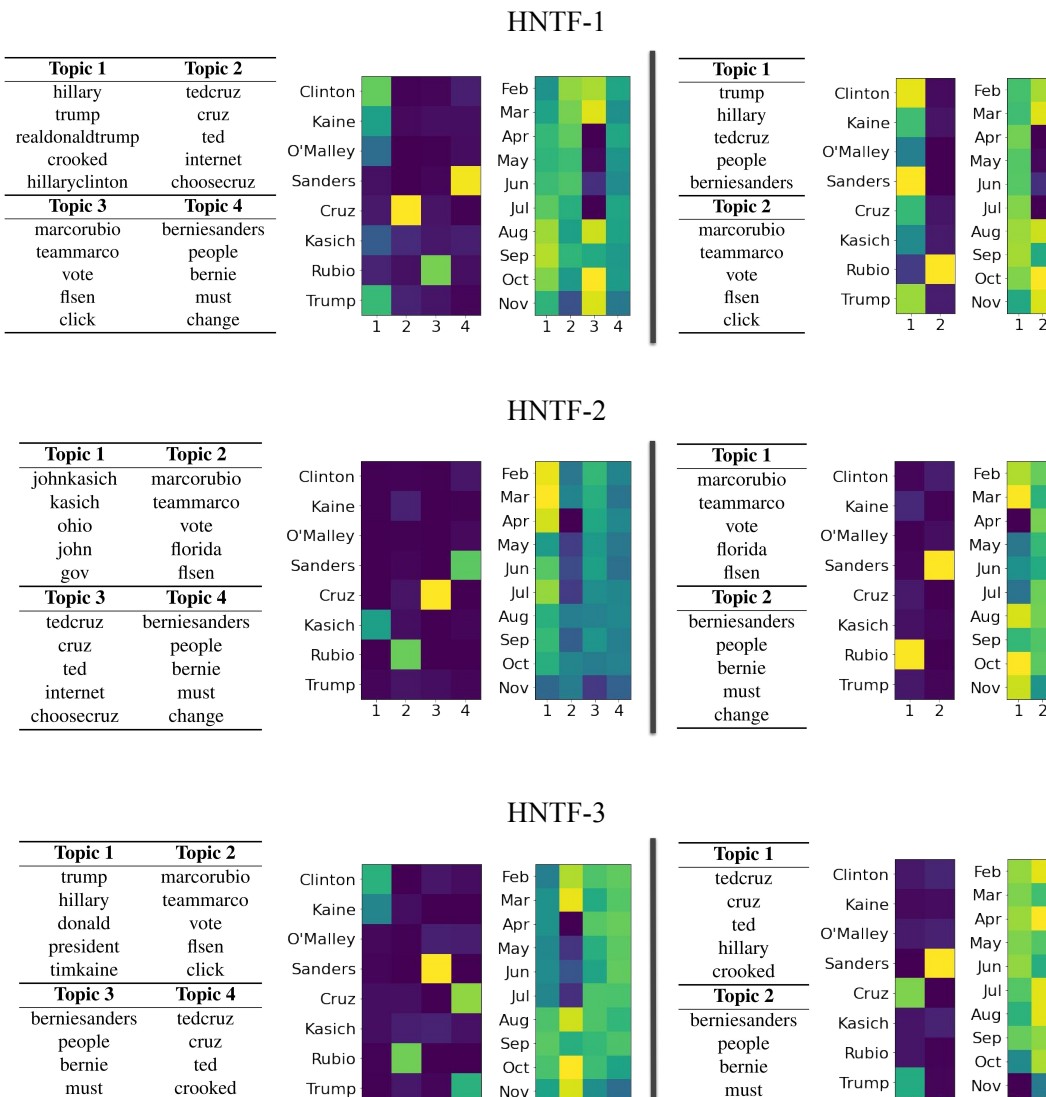

Figure 10: Here we display a three-layer HNTF on the Twitter dataset from Section 3.2 at ranks $r = 8$, $r^{(0)} = 4$, and $r^{(1)} = 2$, run separately for each of the possible ordering on the data tensor. We display the top keywords and heatmaps of topics in the candidate and temporal modes at ranks 4 (left) and 2 (right). We note that the rank 8 factorization is identical to that of Neural NCPD, so we do not re-display it here (see Section 3.2).

In Figure 10, we display the results from running HNTF on the Twitter dataset in Section 3.2, excluding the topics at rank 8 because they are identical to those learned by Neural NCPD (see Section 3.2). We see that while the factorization for the first possible ordering is similar to that of Neural NCPD and contains significant meaningful topic modeling information, the other two orderings lose significant information by the last layer and, and have topic presence and from only 2 or 3 of the eight candidates.

VIDEO DATA EXPERIMENT

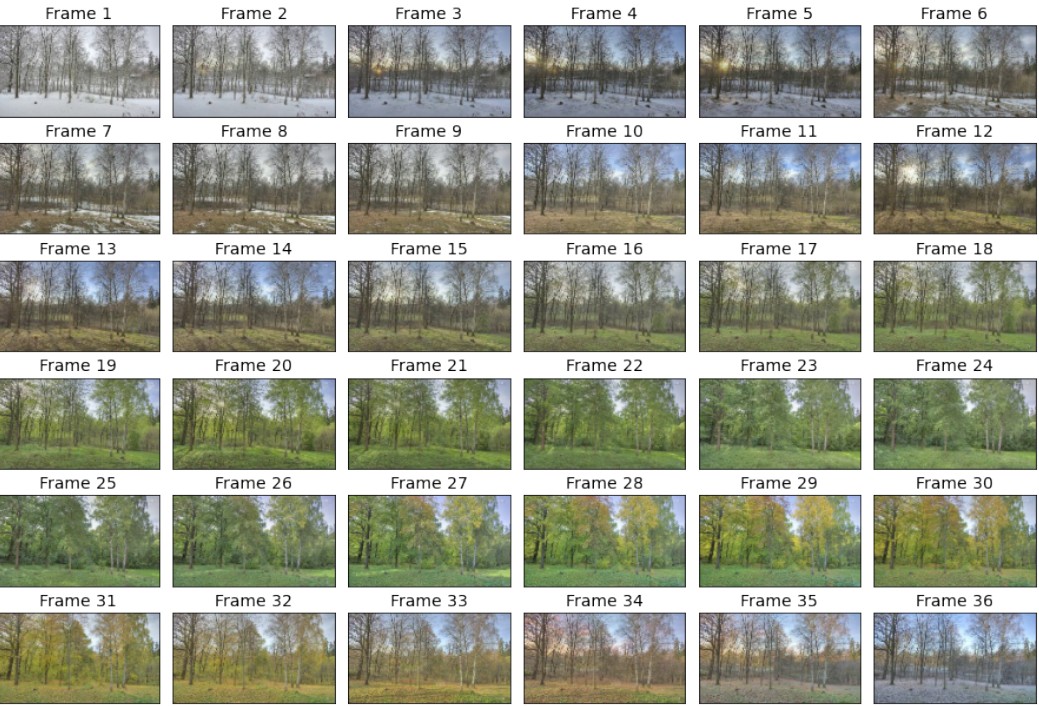

Figure 11: Here we display the first 36 of 37 frames of the time lapse video dataset from Section 3.3 (The 37th frames is included in Figure 6

In Figure 11, we display the first 36 of 37 frames of the time lapse video dataset from Section 3.3 (the 37th frame is included in Figure 6) in order to make it clear how seasons progress throughout the frames. We see that the video begins in the white winter months, transitions to spring at around frame 16, and stays green until it transitions to fall around frame 28.

In Figure 12, we display the $\boldsymbol{S}_3^{(0)}$ matrix (top) and $\boldsymbol{S}_3^{(1)}$ matrix (bottom) produced by Neural NCPD on the time-lapse video tensor described in Section 3.3. By examining the $\boldsymbol{S}$ matrices from our Neural NCPD algorithm, we are also able to see the hierarchical relationship between the topics from different ranks. In the $\boldsymbol{S}_3^{(0)}$ matrix, we see the hierarchical relationship between the rank 6 and rank 8 topics. In the $\boldsymbol{S}_3^{(1)}$ matrix, we see the hierarchical relationship between the rank 3 and rank 8 topics. We note that the $\boldsymbol{S}_3^{(0)}$ matrix (top) illustrates that topic one of rank 6 NCPD is closely related to topic eight of rank 8 NCPD, and $\boldsymbol{S}_3^{(1)}$ (bottom) similarly illustrates that topic two of rank 3 NCPD is closely related to topic eight in rank 8 NCPD; these relationships are unsurprising because, as seen in Figure 7 in the main text, these topics are present temporally during winter and fall and spatially in the sky behind the trees.

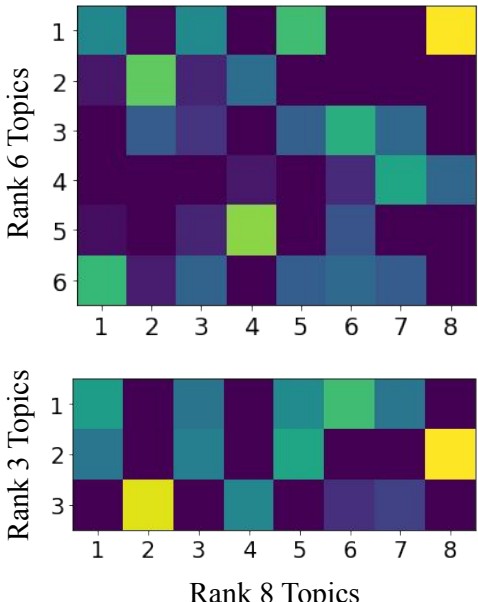

Figure 12: The $\boldsymbol{S}_3^{(0)}$ matrix (top) and $\boldsymbol{S}_3^{(1)}$ matrix (bottom) produced by Neural NCPD on the time-lapse video tensor described in Section 3.3.

