# OpenReview forum: "Neural Nonnegative CP Decomposition for Hierarchical Tensor Analysis"
_ICLR.cc/2021/Conference — Reject_

### Official Review · AnonReviewer1 · 2020-10-28
**A short review for Neural Nonnegative CP Decomposition for Hierarchical Tensor Analysis**

**Rating:** 4
**Confidence:** 2

**Review:**

Short summary:

The paper introduces a promising new method, hierarchical nonnegative CP decomposition (HNCPD), as well as a training method for the HNCPD, neural NCPD, for topic modeling problems. After presenting some fundamentals on the topic, as well as some related work, a detailed and formal mathematical description of the suggested method is given. Finally, results of experiments on three data sets (synthetic data, image data and text data) are utilized to evaluate the suggested methods.

Potential Improvements:
- Equation (4) introduces the forward propagation for a NNMF, which is crucial for the HNCPD and Neural NCPD. However, the method of determining

$\argmin\ limits_{\boldsymbol{S}}$

 is not introduced. Further, it is not shown how the derivative of the argmin function with respect to

$\boldsymbol{A}^{(l)}$

 is formed, which is later on used for the backpropagtion.

- The introduction mentions the importance of topic modeling given large amounts of data, as well as multi-modal tensor data. However, the present experiments only utilise rather small uni-modal data sets (for instance a selection of 37 forest images). It would therefore be of value to answer 1) how the experiments would scale to higher dimensions (for instance for images of the forest time laps) and 2) how the suggested approach would work on a multi-modal database.

- The analysis of experiments in section 3.2 and 3.3 is rather qualitatively than quantitatively. A quantitative measure like the reconstruction loss would most likely be of value.- Sections 3.2 and 3.3 also lack a comparison to other methods.

- The conclusions section could include a short summary of future work on this novel approach.


Minor Comments:
- The for-loop over the two variables i and l in Algorithm 1 could be expressed as two for-loops to be more precise.
- In section 3: presidential election --> United States presidential election
- In section 3: combines the the rank 8 --> combines the rank 8
- In section 3: we display the the results --> we display the results

Overall Evaluation:
The paper is well and clearly written, the significance of this work, however, needs to be further proven by experiments. The originality seems moderate, as already existing concepts of neural NMF and Hierarchical NMF are applied to NCPD.

---

> ### Author Response · Authors · 2020-11-19
> **Response to AnonReviewer1**
>
> We thank the reviewer for their thoughtful comments.  In particular, we appreciate the focus on improving our experimental section by adding comparisons (quantitative and qualitative) to other models.
>
> Responses to Potential Improvements:
>
> - We have added details for how q(A,S) is computed in the Neural NMF section and we added a citation for the derivative of q(A,S) in the backpropagation section.
>
> - We thank the reviewer for this comment.  Note that the Twitter political dataset has one mode with dimension 12,721, so we consider this dataset to be of relatively large size.  Our experiments include three-mode tensor data since that was the most natural representation, and was easiest to visualize.  Our model and methods scale easily to tensors with more modes, however.
>
> - We thank the reviewer for this comment. In topic modeling, qualitative analysis is often more important than quantitative analysis, as it is possible to create a model that produces a very accurate approximation to a data set while offering little topic modeling information. In the case of HCNPD vs. NCPD, it is expected that directly performing an NCPD at each desired layer will produce better reconstruction loss, as the topics at each layer are no longer constrained to be a linear combination of the topics at the previous layer.  Thus, in these experiments we present qualitative analysis and illustration of the presence of a clear hierarchical topic structure.
> We have added the reconstruction loss to our experiments. We also added comparisons to another method, Cichocki et al. (2007a), as well as an analysis of the difference between our models. For the reasons above, we still focus our analysis on qualitative analysis, but we agree that these quantitative measurements are a useful and important addition for readers interested in reconstruction performance and the use of the HNCPD model for tasks other than topic modelling.
>
> - While we agree that discussion of future directions would be valuable, we are not able to include due to space constraints.
>
> Responses to minor comments: We have addressed all, and have fixed the for-loop representation of Algorithm 1.

---

### Official Review · AnonReviewer6 · 2020-11-05
**Interesting paper which could be improved if certain unclear parts are explained better**

**Rating:** 6
**Confidence:** 4

**Review:**

SUMMARY:

This paper presents a hierarchical nonnegative CP tensor decomposition method. It also proposes a training method that leverages forward and backward propagation. The method is tested on both synthetic and real datasets. These experiments illustrate how the method can be used to discover topics and how they vary over time. The hierarchical nature of the method makes it possible to group the topics into supertopics in multiple steps.

I thinks the paper is well-written for the most part. The topic is interesting and should be of interest to the ICLR community. It seems like the authors combine existing ideas (hierarchical and neural NMF, NCPD) into a new method. While this may be somewhat incremental, it seems like the authors are doing something that hasn't been done before. My main issue with the paper is that there are a few portions that are currently difficult to follow. In particular, in the discussion on approximation in Section 2.1, it is not clear if this idea is used in the implementation, which might make it difficult to replicate the results. Fixing this would help improve the quality of the paper. I provide more details on which parts are unclear below.


ADVANTAGES:

- Introduction provides a clear overview of what has been done before. It is easy to see where the present paper fits in with the existing literature. It also does a good job of connecting to applications.

- The experiments in Sections 3.2 and 3.3 are convincing and nicely explained, especially the illustrations of how topics vary over time. The last sentence in Section 3.3 help explain the benefit of a hierarchical method.

- Well-written for the most part.

- Should be of interest to ICLR community.


CONCERNS/QUESTIONS:

- In the last sentence on page 2, it is not clear which loss function is used for Neural NMF when doing the backpropagation. Is it the expression in the Frobenius norm in the paragraph titled "Hierarchical NMF (HNMF)" above?

- In Section 2.1, I follow the discussion up to Equation (6). But the rest of the first paragraph is difficult to follow. For example, it's not clear to me how the columns of the hierarchical NMF factors are used to form NCPDs of ranks $r^{(0)}$, $r^{(1)}$, ..., $r^{(L-2)}$. The subsequent discussion about the dependencies and indices is also unclear. Perhaps this discussion could be made more clear with the help of an example?

- In Section 2.1, I think the second paragraph is clear until Equation (7), but the rest of the paragraph is difficult to follow. The discussion about approximating the relationship between the columns of $A_i^{(0)}$ and $A_k^{(0)}$ is unclear. What do you mean by "relationship"? In what sense is $(W_i)_{p_1, p_2}$ approximating it? It is not clear why the definition in Equation (9) is chosen. Is the idea that $[[ \tilde{A}_1^{(0)}, \ldots, \tilde{A}_k^{(0)} ]] \approx [[ \tilde{X}_1, \ldots, \tilde{X}_k$ ]]? In the experiments, do you use this approximation, or do you use the NCPD combined with HNMF for each factor matrix as discussed in the beginning of Section 2.1?

- In the last sentence of Section 2.1, you mention that "Neural NCPD allows factor matrices for all other modes to influence the factorization of a given mode." It makes sense that this is the case when all parameters, including the initial NCPD, are trained together via backpropagation. But does this still remain true when the initial NCPD is computed independently at first and then kept fixed throughout the fitting of the HNMFs, like in Algorithm 2? As far as I can tell, the approach in Algorithm 2 of keeping the initial NCPD fixed through the training is also what you do in the experiments.

- In the experiments, do you use a combination of the loss functions in Equations (11) and (13) (e.g., C+E), or just one of them? This is not clear from the discussion. Also, since the NCPD is computed by itself at the start of the algorithm and then kept fixed, isn't the term $|| X - [[ X_1, \ldots, X_k ]] ||_F$ in (13) a constant that could be ignored?

- For the discussion in Section 2.2 about the derivatives, it would be a good idea to let the reader know that there is a more in-depth explanation available in the appendix.

- It is not clear how the Standard NCPD in Section 3 is computed. Could you perhaps add an explanation in the appendix or point to a relevant reference?

- Below Figure 2, you say that $g \sim N(z; 0, \sigma^2)$. What does $z$ here mean?

- For the experiment in Section 3.2, did you try keeping each frame as a matrix in the tensor? In other words, did you try reshaping the tensor into a 4-dimensional tensor of size 37 $\times$ 3 $\times$ (number of x pixels) $\times$ (number of y pixels)? One motivation for using tensors is to avoid having to vectorize things like images, so it would be interesting to know if this also worked well. I don't think you need to change the current example, I'm just asking out of curiosity.


MINOR CONCERNS/QUESTIONS:

- In Figure 1, the colors are a nice addition, but they're quite muted which makes it hard to see them. Could they be made brighter? Also, should the red line along the columns of the $S_2^{(0)}$ matrix instead be along the rows of the $A_2^{(0)}$ matrix?

- In Figure 1, in the caption, the tensor $X$ is not using the bold tensor notation.

- Below Equation (7), in the definition of $\alpha_{j_1, j_2, \ldots, j_k}$, should the sum go to $r$ instead of $r^{(0)}$? It looks like it should based on the derivation in the appendix.

- Equation (15) should end with a comma instead of a period since the sentence keeps going below.

- In the second sentence of Section 3.1, the word "size" appears twice in a row.

- At the bottom of page 7, should the tensor size be 8 x 100 x 12721 instead of 8 x 10 x 12721 since the number of tweets are capped at 100?

- In the text to the left of Figure 7, the words "the" appears twice in a row in two places.

- In the appendix, in the section "HNCPD expansion", in the 2nd sentence (starting with "We have that by definition..."), I think the square bracket "]" should be removed on the right hand side of the equation?

#######################

Update:

I thank the reviewers for their responses. I appreciate the effort they put into clarifying the paper. However, I still think Section 2.1 in particular is difficult to follow. I will therefore keep my original rating.

---

> ### Author Response · Authors · 2020-11-19
> **Response to AnonReviewer6**
>
> We thank the reviewer for their thoughtful comments, and in particular, for their detailed questions regarding Section 2.1, which encouraged us to significantly clarify this section.
>
> Responses to Concerns/Questions:
>
> - This is written for an arbitrary cost function, we don’t specify the possible cost functions here due to space constraints, but specify which cost functions are utilized in the Neural NCPD section, which are used to update each Neural NMF branch.
>
> - We thank the reviewer for this comment and agree that this approximation was not clear previously.  We have reordered much of this section, to make more clear why the approximation is necessary, and then how the approximation is computed.  We have also added several sentences to the paragraph containing equations (8) and (9).  In particular, we have added explanation that the NCPD of rank r^(0) at the second layer of HNCPD is [[\widetilde{\mA}_1^{(0)}, \cdots, \widetilde{\mA}_k^{(0)}]].
>
> - We have reworded this section more carefully.  In particular, we have provided an intuitive description of the W_i matrices, which approximate many different-index vector outer product terms in (7).  We have clarified how these approximations are defined, specifically that the W_i matrices transform the A_i(0) to ~A_i(0) by collecting the approximate contribution of all columns of A_i(0) in vector outer products with A_k(0)_{:,p_2} into ~A_k(0)_{:,p_2}.  We use this approximation scheme for the Video and Twitter experiments, where the relationships between the factor matrices are relevant for interpretability. We added a sentence to the beginning of Section 3 to make this detail clear.
>
> - We thank the reviewer as we had not clarified this subtlety sufficiently.  Our backpropagation does not update the factor matrices X_1, …, X_k in the first (NCPD) layer, but the backpropagation update for matrices A_i^(0), …,  A_i^(L-2) is influenced by all factor matrices A_j^(l), since our loss functions incorporates all factor matrices. Even when X_1, …, X_k are fixed, we do not calculate each HNMF branch separately, but rather learn all of them simultaneously utilizing a loss that depends upon the factorization in each branch.
>
> - We thank the reviewer for this comment, as we had not made clear this aspect of our experimental setup. We use energy loss for all of our experiments, and this first term can indeed be ignored when we keep the initial factor matrices fixed. We have added additional details to Section 3 to make this clear.
>
> - We have added a sentence clarifying this before Section 3.
>
> - We have added a sentence with these details immediately before Section 3.1.
>
> - Thank you for catching this typo, we have fixed it.
>
> - We thank the reviewer for this interesting question.  If we represent each frame as a two-mode slice of the tensor and produce a 4-dimensional tensor as you describe, then an NCPD decomposition would produce topics which are represented by rank-one matrices in two modes. Thus, these topics would give general spatial presence of colors and times, but not interesting shapes, such as trees or clouds. This agrees with results of [arXiv:2009.07612 (2020)], where the authors apply NCPD to extract image patches, and find that when the image is not flattened, the patches have constant color and their shape varies only by horizontal and vertical lines.
>
> Minor concerns/questions: We have addressed all.  Regarding the Twitter dataset tensor construction, the mode with dimension 10 represents months. For a given month i and candidate j, all the tweets for that month are aggregated (via bag-of-words) into a single vector of length 12721, which is the value of each fiber X_{i,j,:} for tensor X.

---

> > ### Comment · AnonReviewer6 · 2020-11-21
> > **Additional questions**
> >
> > Thank you for your response and clarifications.
> >
> > I'm still confused by Section 2.1 and when the described approximation is used. My questions below are for manuscript_red.pdf.
> >
> > For an $\ell$-layer network, as far as I can tell, the purpose of Alg. 1 and Alg. 2 is to fit the tensor in Eq. (11) to the input tensor $X$. What is unclear is what the point of the approximation idea in Section 2.1 is. Is this part of the model, so that the $W_i$ matrices and $\tilde{A}_i^{(j)}$ matrices are also fitted during the training process, or is the approximation in Section 2.1 done _after_ the model in  Eq. (11) has been fitted? If it's the latter, is the point of this to help make interpretation/analysis easier?
> >
> > Is the model in Eq. (11) what you refer to as HNCPD? Based on the title of Section 2.1, it seems like HNCPD also incorporates the suggested approximation. Is the approximation an optional extra step that can be used to improve HNCPD?
> >
> > I hope this helps clarify what my confusion is.

---

> > > ### Author Response · Authors · 2020-11-21
> > > **Response to additional questions**
> > >
> > > Thank you for clarifying your previous questions.  With this clarification, we have better revised Section 2.1.  Following the training process provided in Alg. 1 and Alg. 2, the resulting HNCPD approximation has a hierarchical structure that depends on all of the S matrices, which makes interpretability and visualization difficult. As you suggested, we apply our approximation scheme to the factor matrices output by this learning process for interpretability purposes, so this step is independent from the training process. To clarify this, we rephrased the introduction of the approximation scheme in Section 2.1 to make the purpose and application of this approximation clearer.  In particular, we added an equation (7) which explicitly states the layers of the HNCPD, and states the purpose of the following approximation scheme. We additionally removed some language that was unclear regarding the purpose of the approximation scheme.  We also added a sentence to the beginning of Section 2.2 and to Section 3 clarifying how we use the approximation scheme.

---

### Official Review · AnonReviewer5 · 2020-11-06
**Good work but the quality is not enough**

**Rating:** 4
**Confidence:** 4

**Review:**

Summary:
In this paper, an extension of nonnegative CP decomposition called hierarchical nonnegative CP decomposition (HNCPD) is proposed. This method is designed to capture the hierarchical structure in e.g., topic modeling. Also, an optimization method called neural NCPD is proposed. Several decomposition results of synthetic data, video data, and Twitter data are presented.

Strong points:
- S1. A new algorithm is proposed.
- S2. Experimental results with real data are reported.

Weak points:
- W1. The technical contributions are small.
- W2. The motivation of HNCPD is unclear.
- W3. The paper is not clearly written.

My recommendation:
Although this paper contains some interesting ideas, I feel the overall quality is not high enough to be accepted in the ICLR community. Here I would like to elaborate on the reasons.

W1. The formulation of HNCPD is not new. Cichocki et al. (2007) proposed a more general form of HNCPD; see Eq. (13) of the following paper. So the technical contribution of this paper is summarized to the development of neural NCPD, but it is a direct extension of Gao et al. (2019).
````
Cichocki A., Zdunek R., Amari S. (2007) Hierarchical ALS Algorithms for Nonnegative Matrix and 3D Tensor Factorization. In: Davies M.E., James C.J., Abdallah S.A., Plumbley M.D. (eds) Independent Component Analysis and Signal Separation. ICA 2007. Lecture Notes in Computer Science
```````
W2. It is not clearly explained how HNCPD is beneficial to real-world applications. For example, the pictures in e.g. Fig. 4 seem to tell us some interesting information, but how can we use that? In topic modeling, what kind of knowledge/insights can we get from the results? And also, how can we quantify the benefits?

W3. Some part is not easy to read.
- In Section 2, $r$ is defined as the number of super topics. However, in Table 1, $r$ seems to be used as the number of subtopics, which is previously defined as $r_0, r_1, r_2$. A similar inconsistency is observed in Fig4, Fig6, etc.
- The interpretation of experimental results is unclear. In Section 3.2, it is claimed that HNCPD is better than NMF because "the chromatic NMFs obscure much of the chromatic interaction". Here, the true structure of chromatic interaction is hidden and we cannot observe it. So in what sense can we show the superiority of HNCPD?

---

> ### Author Response · Authors · 2020-11-19
> **Response to AnonReviewer5**
>
> We thank the reviewer for their thoughtful comments and thorough analysis of our paper, and for taking the time to highlight an important model comparison that was lacking from our previous draft.
>
> **W1:** We respectfully disagree with the idea that the formulation of HNCPD in Cichocki et al. (2007) is a more general form of HNCPD than the one we propose. We note especially that in Cichocki et al. at each step of HNCPD, each layer of the next level of factorization is decided by two of the factor matrices (D and S) while the third factor matrix, A, is held constant. This significantly weakens the generality of the factorization as one of the three modes is not used in deciding the hierarchy, which undermines the factorization and topic modelling capability when the factor matrix containing the most important hierarchical structure is in the 1st mode. We added a new experiment to the papar (see Table 2, Appendix Figure 10) which demonstrates that the factorization is weakened when in the Twitter experiment, the first mode is the word corpus, leading to significant increases in reconstruction loss and worse topics. By contrast, in our model all three factor matrices are used symmetrically during the factorization. We have added a comment describing this to the first paragraph in the “Contributions” section and to the “Other Related Work” section.
>
> We agree that our model and optimization method are an extension of Gao et al. (2019) but we do not believe that it is a direct or simple extension. For example, we note that during the backpropagation step, the updates for each NMF “branch” are not simply the derivatives of a Neural NMF layer, because the branches are not independent, so we must backpropagate to all branches simultaneously and consider the factor matrices along all branches when updating any single factor matrix in a given branch. We have added a comment to this point to the second paragraph of the “Contributions” section.
>
>
> **W2:** When designing the experiments, we intended for the Twitter experiment to showcase the value of learning hierarchical topic relationships, while the video experiment was intended to showcase the topic modeling value offered by NCPD over NMF. Thus, in the video experiment, the desired take-away is that in our 3-mode data set (spatial, temporal, chromatic), the topics learned from NMF by slices will lose information from at least one of these modes. In the Twitter experiment, HNCPD offers additional interpretability that cannot be gained from performing a single NCPD at a given rank. At rank 8, one would see the keywords from each candidate, but not the relationships between candidates. At rank 4 and 2, one could see which candidates are grouped together, but not why they are grouped together. On the other hand, with HNCPD we can gain such interpretability; for example, we see Cruz’s and Kasich’s rank 8 topics combine at rank 4 due to their similar temporal presence, and we see Trump’s and Clinton’s rank 8 topics combine at rank 4 due to their similar keywords.
>
> In order to improve the clarity on these points, we made two main changes. First, we changed the order of experiments so the Twitter experiment appears before the video experiment, as the Twitter experiment is better able to demonstrate the importance of hierarchy. Second, we moved the S matrices, which illustrate the relationships topic learned at different layers, out of the appendix and into the main body. These matrices demonstrate how topics at each rank relate to the topics at the original rank, which helps to elucidate the hierarchy.
>
> We believe that in general, the tasks of topic modeling for imaging applications are both interesting and important. We chose to use the simple forest time lapse data set because there is a clear intuition for the expected topics, which allows for an intuitive analysis of the experimental results. One possible application of such an experiment is to use the learned spatial topics and their temporal presence to visualize properties of the frames that are most indicative of a given season; or more generally to visualize the spatial or chromatic features of frames that are most correlated with a given temporal range. Other possible applications include image segmentation and image classification.

---

> ### Author Response · Authors · 2020-11-19
> **Response to AnonReviewer5 (part 2)**
>
> **W3:** Thank you for catching the typo regarding use of r and r_0, r_1, etc.  We have corrected the instances of r to r0, r1, etc.  We have also clarified these labels in the figures.
>
> To address the concern of the chromatic interaction being hidden, we added the remaining frames of the video data set to the appendix so that the evolution of seasonal changes and corresponding chromatic changes is clearer.  We expect the chromatic presence of our topics to match the expected colors of the season (e.g., we expect a topic in the Fall with spatial presence in the trees to have red). We specifically chose a data set in which we know what kinds of topic modelling information to expect in each mode (in accordance with the common understanding of the seasons). We felt the inclusion of this experiment would provide intuition to readers for the value of such a topic modeling technique.
>
> We note that the topics learned by HNCPD align with the expected topic modelling information (such as topics during spring time having green leaves). Meanwhile, the topics produced in NMF by slices did not provide any information about chromatic structure of topics, as slices taken from each of the three base colors produced nearly identical spatial and temporal topics.

---

### Author Response · Authors · 2020-11-19
**Response to Reviewers**

We thank the reviewers of our paper "Neural Nonnegative CP Decomposition for Hierarchical Tensor Analysis" for their thorough and thoughtful comments.  In response to these comments, we have made several revisions to the paper, and believe they have significantly improved our manuscript.  We summarize next the most major changes to the paper, but we also respond individually to the reviewers highlighting the changes made in response to their comments.  To assist with reviewing these changes, we have included in the supplementary material a PDF titled "manuscript_red.pdf" in which all changes are highlighted in red.

Summary of revisions:
1) We have added a contributions section which addresses the concerns regarding novelty of both the hierarchical NCPD model (in comparison to Cichocki et. al. (2007a), referred to as HNTF), and the novelty of the Neural NCPD model (in comparison to Gao et. al.).  We highlight that our contributions are not simple incremental advances beyond either.

2) We have also addressed the concern regarding novelty of HNCPD with discussion of Cichocki et. al (2007a) in the "Other Related Works" section, and have added comparisons to this model in the experimental section.  We highlight the disadvantage suffered by Cichocki et. al. (2007a), that the model is dependent upon the chosen representation of the data (specifically the ordering of the tensor modes).  Our model does not suffer from this disadvantage as all tensor modes are treated symmetrically during learning.

3) We have significantly clarified Section 2.1 which was previously unclear.  We illustrate clearly how and why we form our NCPD approximation at later layers.  In particular, we removed ambiguous language describing the motivation and have replaced with both an intuitive and rigorous description of the approximation.

4) We have improved the description of the backpropagation procedure and indicated how it builds upon Gao et. al. but does not immediately follow.  We have added a citation to the references containing necessary partial derivative calculations, and have highlighted how we combine these in Neural NCPD and allow the Neural NMF branches to interact.

5) We have better illustrated the importance of our experiments, highlighting how to access the topic hierarchy, and highlighting the improvement over standard NCPD and standard HNCPD.  We have moved a figure from the appendix to the Twitter experiment section to better visualize the learned hierarchy.  We additionally reordered the experiments to ensure reader focus on the important benefits offered by Neural NCPD.

6) We have added additional description to experimental set-up (indicating the cost-function, the architecture of Neural NCPD, and how the approximation scheme is used), clarifying previous ambiguities.

7) We have added additional comparisons to the experimental section, specifically comparing to Cichocki et. al. in the synthetic experiment and adding comparisons to all other NCPD models for the Twitter experiment.  We have added quantitative measures to all experiments, recording the relative reconstruction loss.

---

> ### Author Response · Authors · 2020-11-21
> **Updated submission**
>
> In light of AnonReviewer6's response, we have updated our submission.  We include in our supplementary material the previously revised manuscript (version 2) with changes in red as "manuscript_red.pdf" and the newly revised manuscript (version 3) with most recent revisions in red as "manuscript_red2.pdf".

---

### Decision · Program_Chairs · 2021-01-07
**Final Decision**

**Decision:**

Reject

**Comment:**

The paper presents a hierarchical version of NMF for the CP decomposition of tensors.

The idea is similar to Chinocki etal 2007 and extends Gao etal 2019, and in Chinocki was presented for the standard linear formulation with regularisation terms.  The extension here doesn't use the standard ALS algorithm but rather presents a neural network analogue, though the functions are still linear, its just that back-prop etc. are used for the computation.  The authors point out their formulation is a more flexible representation and optimisation (in response to AnonReviewer5), and thus represents an improvement.  While this is an interesting implementation, in NNs, the model is still fairly simple.

Moreover the experimental results are restricted to a few data sets.  There are literally hundreds of NMF variants in publication and many different evaluations are done.  The experimental work here, while showcasing the work, is not extensive.  For instance, more empirical comparisons should have been made against prior hierarchical NMF on a battery of data.

So this is good, publishable work, and the authors have repaired many of the issues raised by the reviewers.  The work, however, is borderline in empirical work and the contribution is not strong.